# Determining the current size and state of subvolcanic magma reservoirs

Gregor Weber [1✉], Luca Caricchi [1], José L. Arce [2] & Axel K. Schmitt[3]

Determining the state of magma reservoirs is essential to mitigate volcanic hazards. However, geophysical methods lack the spatial resolution to quantify the volume of eruptible magma present in the system, and the study of the eruptive history of a volcano does not constrain the current state of the magma reservoir. Here, we apply a novel approach to Nevado de Toluca volcano (Mexico) to tightly constrain the rate of magma input and accumulation in the subvolcanic reservoir. We show that only a few percent of the supplied magma erupted and a melt volume of up to 350 km$^3$ is currently stored under the volcano. If magma input resumes, the volcano can reawake from multi-millennial dormancy within a few years and produce a large eruption, due to the thermal maturity of the system. Our approach is widely applicable and provides essential quantitative information to better assess the state and hazard potential of volcanoes.

[1] Department of Earth Sciences, University of Geneva, Rue des Maraîchers 13, CH-1205 Geneva, Switzerland. [2] Instituto de Geología, Universidad Nacional Autónoma de México, Coyoacán, 04360 Ciudad de México, México. [3] Institut für Geowissenschaften, Universität Heidelberg, Im Neuenheimer Feld 235, D-69120 Heidelberg, Deutschland. ✉email: gregor.weber@unige.ch

Understanding the temporal evolution, size and physico-chemical state of igneous plumbing systems beneath volcanoes is of paramount importance to develop quantitative prognostic models for volcanic activity[1]. It is now well established that most volcano plumbing systems are built over prolonged periods of time by pulsed magma injection[2–5] and that magma reservoirs are complex with transiently interconnected and vertically extensive storage regions in the Earth's crust[6–8]. To unravel such complexity and quantify parameters such as the volume of potentially eruptible magma, which are essential for hazard assessment, we rely on geophysical methods and the record of past eruptions.

Geophysical surveys typically estimate melt fractions <20% in crustal reservoirs beneath active volcanoes[9,10], which is significantly lower than the crystallinity of almost all observed erupted products[11,12]. However, geophysical estimates represent average melt fractions because of the limited spatial resolutions of such techniques[13,14]. In contrast, geological and geochemical data provide information on the structure, dynamics, and timescales operating within volcanic plumbing systems[1,5,8,14–16]. Zircon crystals have proven particularly useful to trace the temporal evolution and physico-chemical state of crustal magma bodies[17], as crystallization age spectra and trace element geochemistry of igneous zircon can be used to constrain the thermal histories of magmatic systems[18–23]. Additionally, diffusion chronometry on chemical gradients in various mineral phases has been used to gain valuable insights into the timescales of pre-eruptive magmatic processes (e.g. ref. [16] and references therein). However, while the analysis of geological and geochemical records provides quantitative information on the processes and timescales associated with past eruptive activity, these data are disconnected from the current status of a magmatic system.

To constrain the current status of a magmatic system and eventually forecast its future evolution, different approaches have been proposed. The rationale behind these approaches is to integrate geological, geochemical, and/or geophysical records with the results of thermal modeling to identify fundamental parameters controlling the temporal evolution of magmatic systems[24–30]. The comparison between models highlights that the thermal evolution of plumbing systems, and the volumes of eruptible magma they contain, depends strongly on the average rate of magma input and on the thermal state of the crust[3,31–34]. Variations of the modality of magma injection and of the geometry of the magmatic system, or magma convection play a minor role[3,24,33]. However, quantitative estimates of average rates of magma input are difficult to obtain from the volcanic record as only a fraction of magma injected into the plumbing system is erupted[26,33,35]. Zircon age distributions in combination with thermal modeling have been shown to provide insights into crustal magma fluxes[24–26].

Here, we present a new approach to the study of the thermal evolution of magmatic systems, which provides much higher resolution (factor < 2) on the rate of magma input into the system with respect to existing models (about one order of magnitude for refs. [24,25]). In contrast to previous studies[24–26], which used a single parameter (i.e. shape of zircon age spectra) to compare modeled and natural zircon age populations, we use the 2σ zircon age span, trace element geochemistry, and estimates of erupted volume. The comparison on the base of various independent parameters provides tighter estimates of the average rate of magma input into the subvolcanic magma reservoir than previous models. Matching natural data to modeled age and temperature distributions, we determine crustal magma fluxes and the extrusive:intrusive (E:I) ratio at unprecedented resolution for Nevado de Toluca (Fig. 1), a dormant dacitic stratovolcano with 1.5 Million years history of explosive and effusive eruptions, situated in the densely populated Central Mexican highland[36–39]. With our approach we can assess the size of the present subvolcanic reservoir. Our results show that the reservoir still contains large volumes of magma, and that it could be reactivated in few years by renewed magma supply from depth, which is of utmost importance for volcanic hazard assessment in such a highly populated area.

## Results and discussion

**Zircon ages and geochemistry.** We have analyzed zircon from four eruptions of the younger pyroclastic sequence of Nevado de Toluca (Fig. 1; eruption ages based on $^{14}C$ geochronology[36]): The 12.5 ka Upper Toluca Pumice (UTP), 13.9 ka Middle Toluca Pumice (MTP), a Block and Ash Flow erupted at 31 ka (BAF) and a massive so far undated pyroclastic density current (WQ; Fig. 2). Zircon crystallization ages for all analyzed grains range from 33.1 ka to 910 ka, comprising ~60% of the ca. 1.5 Ma eruptive history of Nevado de Toluca. Samples from all of the four eruptions span a large fraction of the age range resolvable by $^{238}U$-$^{230}Th$ disequilibria dating (<350 ka). The youngest zircon ages are 33.1 ka for UTP (Fig. 2a), 114 ka for MTP (Fig. 2b), and 71.9 ka for BAF (Fig. 2c). These are 21 ka, 100 ka, and 41 ka older than radiocarbon-constrained eruption ages for these events, respectively[36–38]. The youngest zircon age of 116 ka for the WQ sample (Fig. 2d) provides a maximum estimate for the eruption age of this event. Given the differences of 21–100 ka between youngest zircon crystallization and eruption age for the other samples, the age of the WQ eruption is also likely more recent than the youngest zircon age. The fraction of $^{238}U$-$^{230}Th$ ages in secular equilibrium for the four eruptions varies between 0.22 (WQ) and 0.36 (UTP; Supplementary Fig. 5). U-Pb dating significantly extends the maximum observed age for each of the eruptions to 577 ka for the UTP, 653 ka for MTP, 806 ka for the BAF, and 910 ka for the WQ eruption (Fig. 2a–d). The data show mostly continuous and protracted crystallization history, with smaller gaps, which are particularly evident between 78 and 114 ka for the UTP eruption. Generally, the observed range of ages for individual events overlaps and track prolonged zircon crystallization in the plumbing system of Nevado de Toluca (Fig. 2e).

As for the age spectra, zircon trace elements are similar between eruptions (Fig. 3), indicating that the crystals from different eruptions originated in a common magmatic reservoir. Selected trace element abundances and ratios are shown as a function of zircon Ti content, which is a proxy for zircon crystallization temperature assuming invariant activity of $TiO_2$ in the melt[40]. Ti contents range from 1.41 to 20.26 µg/g with an interquartile range (IQR) between 2.92 and 4.86 µg/g. Zr/Hf (IQR: 38.8–44.4) and Th/U (IQR: 0.21–0.34) ratios show a positive relation with Ti content and typically greater scatter at higher values (Fig. 3a, b). Light rare-earth elements (LREE) Ce and Y show no clear relation with Ti and most of the data clusters over IQRs of 6.45–12.52 µg/g and 797.5–1249.4 µg/g, respectively (Fig. 3c, d). In the case of Ce, two trends seem present: One from the central cluster towards high Ce contents at low Ti and the other at low Ce over a range of Ti contents. Similar to Zr/Hf and Th/U ratios, Y contents show more scatter in the data at higher Ti contents. This general pattern is also observed in ratios of middle to heavy rare-earth elements (e.g. Dy/Yb – Fig. 3e), but with a more pronounced positive relationship with Ti. No systematic relation of Ti and $(Eu/Eu^*)_N$ is evident (Fig. 3f), but scattering of the data typically develops towards lower $(Eu/Eu^*)_N$ from a cluster with IQR between 0.30 and 0.37.

**Thermo-chemical modeling.** Selected thermal model results (Fig. 4) illustrate that while the range of temperatures recorded by

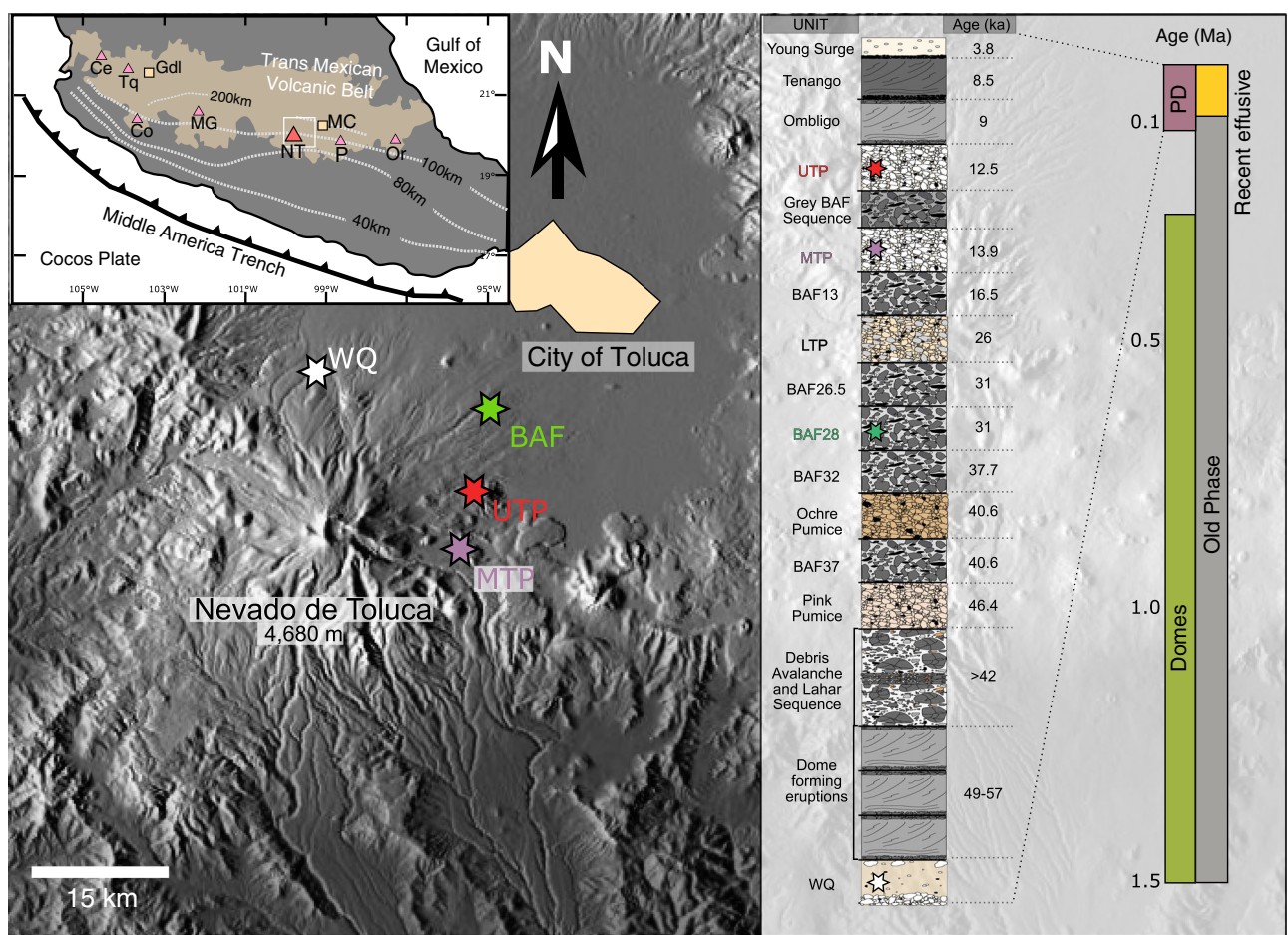

**Fig. 1 Location and studied samples of Nevado de Toluca volcano.** Shaded relief map of the Toluca area is shown with sampling locations of 4 eruptions analyzed in this study marked by colored stars. Red star: Upper Toluca Pumice (UTP), purple: Middle Toluca Pumice (MTP), green: Block and Ash Flow (BAF28) and white: White Quarry Pyroclastic Flow (WQ). The inset on the far right shows the long-term 1.5 Ma eruptive history of Nevado de Toluca and division into eruptive episodes based on $^{40}Ar/^{39}Ar$-geochronology[39]. The young pyroclastic sequence (PD) is shown as composite stratigraphic column, with indicated age relations of the eruptions based on radiocarbon dating[36,37]. The studied eruptions are marked by colored stars in the stratigraphic column. Inset on the top left shows the location of Mexico City (MC) and Guadalajara (Gdl), as well as the position of major volcanic centers (triangles) in the Neogene to recent Trans Mexican Volcanic Belt after ref. [68]. Ce: Ceboruco, Tq: Tequila, Co: Colima, MG: Michoacán-Guanajuato, NT: Nevado de Toluca, P: Popocatépetl, Or: Pico de Orizaba. Dashed lines in the inset indicate the inferred depth of the subduction slab[69].

passive tracers (c.f. Methods section) over time is rather variable, in all models the average temperature typically remains within the range of zircon saturation (e.g. 690–825 °C; Fig. 4). Intrusions built by relatively low crustal magma fluxes (e.g. Figure 4a) show larger time gaps between the first magma injection and the onset of continuous presence of magma compared to reservoirs build by relatively high recharge rates (Fig. 4c). The average temperature in magma reservoirs fed by low (Fig. 4a) to intermediate (Fig. 4b) magma input rates increases progressively over time. At the highest rates of magma input considered here, however, the average temperature increases more rapidly to reach a maximum over relatively short time spans (e.g. 600 ka), but it evolves toward lower average values over time (Fig. 4c). This counterintuitive trend is related to the increasing importance of heat loss to the wall rocks relative to the heat addition by recharge pulses for intrusions that have grown more rapidly to a given size (i.e. surface of contact with the wall rocks). In all simulations, we observe that the initial phase of magmatic activity is characterized by higher variance in average temperatures compared to later stages, where individual recharge pulses have less impact due to the growing volume of the intrusion. Moreover, magma reservoirs built with different rates of magma input will develop time-integrated temperature distributions that differ significantly from each other. Generally, higher magma fluxes result in higher average temperatures (i.e. lower average Zr/Hf or higher Ti in zircon) and greater variance (i.e. larger spread in zircon trace element content) compared to systems assembled at lower magma injection rates (Fig. 4).

Synthetic zircon age distributions calculated (Methods) from the thermal model output illustrating the effect of under-sampling on the modeled zircon spectra are shown in Fig. 5 for a system assembled with a magma flux of $9.1 \times 10^{-6}$ $km^3$ $km^{-2}$ $yr^{-1}$ over 1.5 Ma. In order to assess the effects of under-sampling, we resampled the calculated zircon population. Synthetic zircon age distributions with 300 (Fig. 5a) and 100 observations (Fig. 5b) show a continuous and prolonged crystallization history with the maximum at or close to the end of the thermal simulation. Even for relatively high numbers of sampled zircon ages (e.g., 300), peaks and shoulders appear in the density distribution in different age ranges, thus such features cannot be interpreted geologically as they result exclusively from under-sampling. Importantly, these results show that while the shape of the distribution is difficult to interpret the total spread of zircon age is significantly less affected by under-sampling (Fig. 5).

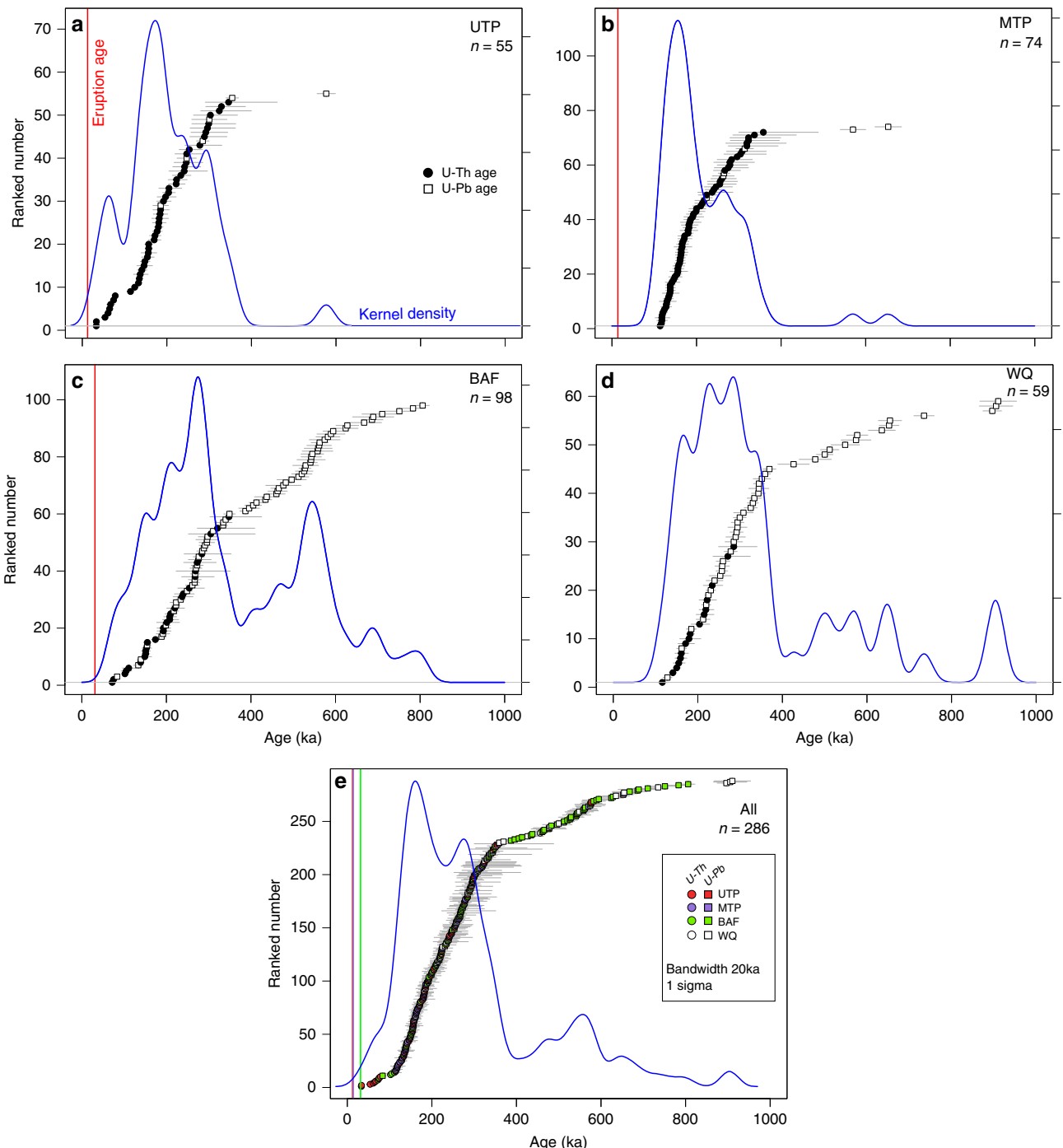

**Fig. 2 Zircon crystallization age spectra from Nevado de Toluca volcano.** Zircon age data are displayed as ranked order plot and kernel density estimate (blue lines) with bandwidth of 20 ka. U-Th ages are shown as filled circles and U-Pb ages as open squares. Eruption ages, if constrained, are marked by vertical red lines[36]. **a** Upper Toluca Pumice (UTP). **b** Middle Toluca Pumice (MTP). **c** Block and Ash Flow 28 (BAF). **d** White Quarry Pyroclastic flow (WQ). Subpanel **e** shows the combined ranked order-age plot for all studied eruptions with color coding for the individual eruptions: UTP: red, MTP: purple, BAF: green, and WQ: white. The number of analyzed spots on zircon crystals (*n*) is indicated on each subpanel and 1 sigma error bars are shown. Combined U-Th and U-Pb zircon age spectra for the individual eruptions are overlapping with each other and indicate prolonged crystallization.

**Zircon age distributions and thermal records**. Zircon age distributions for the investigated eruptions differ in shape and age difference between youngest zircon and eruption ages (Fig. 3). The total duration of zircon crystallization for the BAF and WQ eruptions is longer compared to UTP and MTP, which likely reflects the more limited number of U-Pb ages analyzed for UTP and MTP. Also, the mode of the distribution becomes older with increasing eruption age, which is consistent with zircon

crystallization being continuous over the period over which these eruptions occurred. Gaps in zircon crystallization ages are present between 78 and 114 ka for UTP and between eruption age (13.9 ka) and 114 ka in MTP. Further, a multi-modal age distribution is observed for the BAF eruption. However, as suggested by model curves in Fig. 5, 100 analyses might be insufficient to assign geological significance to zircon crystallization peaks. We therefore suggest that peaks and gaps in zircon age should not be

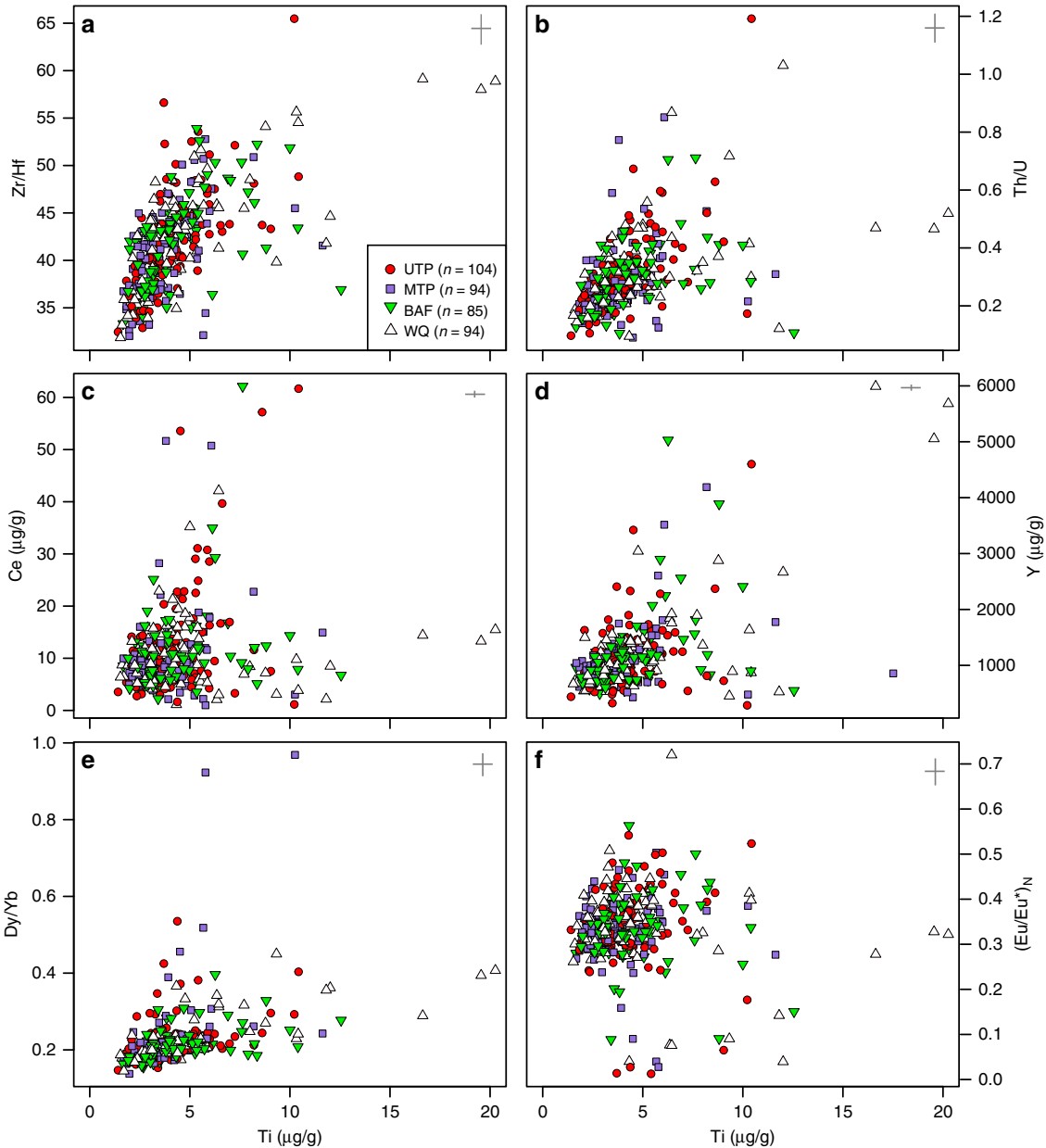

**Fig. 3 Trace element contents and ratios of Nevado de Toluca zircon shown as bivariate plots.** Red circles: Upper Toluca Pumice (UTP), purple squares: Middle Toluca Pumice (MTP), green triangles: Block and Ash Flow 28 (BAF) and white triangles: White Quarry Pyroclastic flow (WQ). The relationship of zircon Ti (µg/g) content with (**a**) Zr/Hf ratio, (**b**) Th/U, (**c**) Ce (µg/g) content, (**d**) Y (µg/g) content, (**e**) Dy/Yb ratio, and (**f**) chondrite normalized[70] europium anomaly $(Eu/Eu^*)_N$ is shown. Note the equivalence of trends and complete overlap of zircon trace element compositions for the different eruptions. 2 sigma errors are displayed on the top right of each diagram. Uncertainties for trace element ratios were estimated by error propagation. The number of spot analyses (*n*) is indicated in the legend for each eruption.

interpreted as resulting from episodicity in magmatic processes without further independent constraints, such as equivalence in $T$–$t$ spectra between different magmatic thermochronometers (e.g. zircon and titanite). Therefore, when comparing measured and calculated distributions of zircon ages, we will focus on the total spread. Additionally, the thermal modeling results show the total age spread is the most reliable parameter varying as a function of the rate of magma input into the plumbing system and the thermal maturity of the crust.

Our model calculations predict that the largest number of zircon ages should be observed close to the eruption age, which is inconsistent with the observed time lags between youngest zircon and eruption ages. Yet, considering that the zircon crystals analyzed in this study have been sectioned through the crystal

cores, younger ages might be underrepresented. In fact, growth of new zircon is energetically favored on already existing crystals[41], and the growth of an equal mass of zircon on a pre-existing crystal leads to the formation of rims the thickness of which decreases with the size of the pre-existing zircon[42]. It is therefore difficult to analyze the youngest growth zone in sectioned zircon, and alternative methods such as depth profiling would be required[43]. Additionally, the youngest zircon growth zones may have been resorbed during thermal rejuvenation of the system, which is consistent with abundant evidence for magma recharge prior to Plinian eruptions at Nevado de Toluca[44]. However, while some rounded grains are present in particular in the UTP sample, the great majority of zircon shows euhedral crystal faces (Supplementary Fig. 1), suggesting that the lack of analyses close

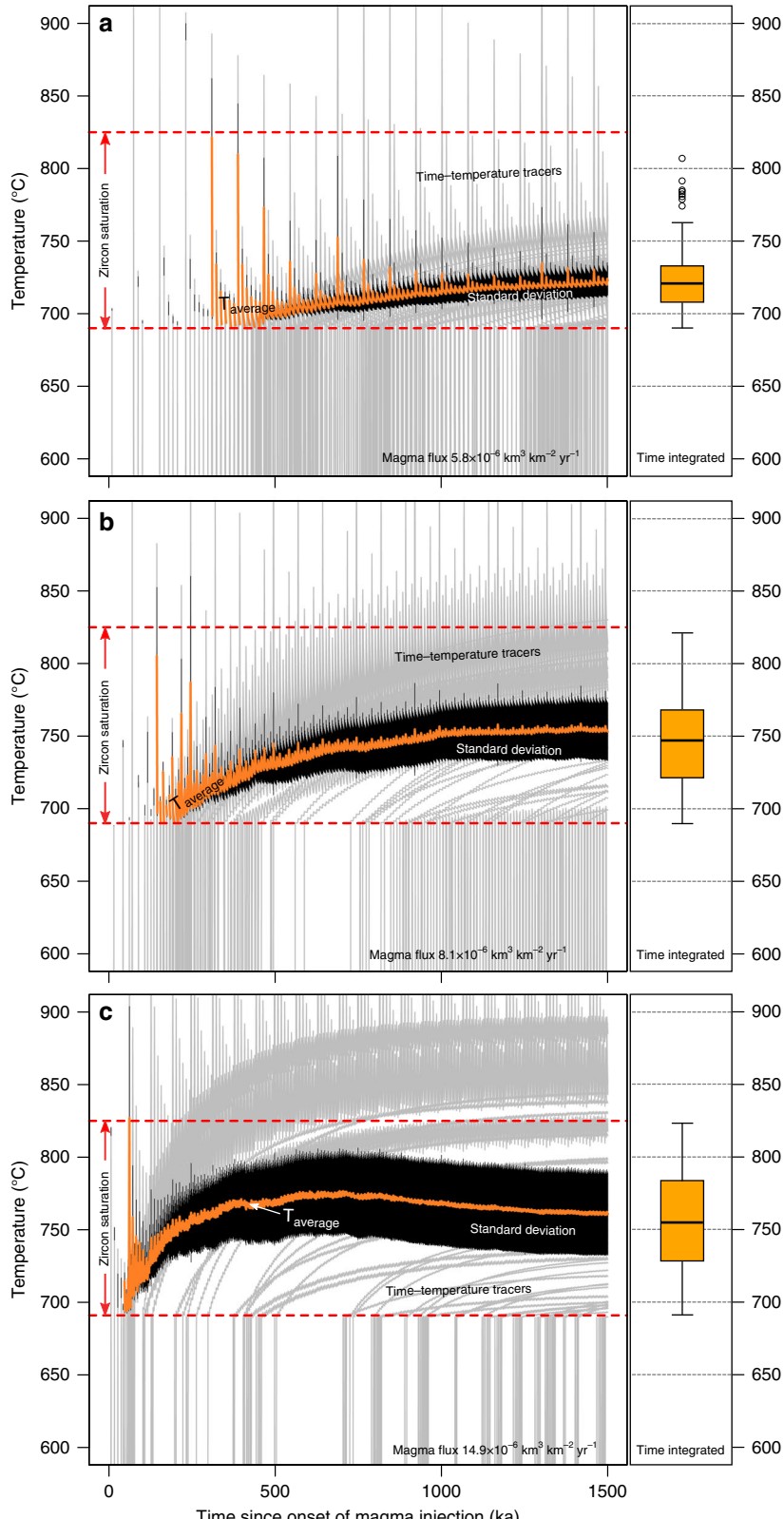

to the eruption ages likely reflects under-sampling of the crystals' outermost rims.

The long history of zircon crystallization with overlapping age distributions for eruptions separated by tens of thousands of years and the consistency of zircon trace element compositions for individual eruptions points towards a relatively chemically homogeneous magmatic system (Figs. 3, 4). These results suggest that the same overall zircon population has been sampled by individual eruptions and can therefore be interpreted to record long-term magma reservoir assembly beneath Nevado de Toluca. We consequently combine the age population (Methods section) for all eruptions to calculate the total age range (2σ range)

**Fig. 4 Time-temperature evolution of modeled magmatic intrusions built by different recharge rates. a** Intrusion built with a magma flux of $5.8 \times 10^{-6}$ $km^3\ km^{-2}\ yr^{-1}$. **b** Reservoir built with a magma flux of $8.1 \times 10^{-6}\ km^3\ km^{-2}\ yr^{-1}$ and (**c**) Intrusion built with magma flux of $14.9 \times 10^{-6}\ km^3\ km^{-2}\ yr^{-1}$. In each subpanel, gray lines show the time-temperature path at a fixed position in the modeled magma reservoir (Time-temperature tracers). Note that only a small subset of the 500 randomly selected $T–t$ paths used in the calculations is shown for the sake of clarity. The apparent high density of gray $T–t$ tracers results from large temperature oscillations that individual tracers close to a magma injection site undergo during their evolution. The average temperature of the magma reservoir is given by the orange curves with 1 standard deviation indicated by black error bars. The temperature interval of zircon saturation (690–825 °C) used in the modeling is marked by red dashed curves. Boxplots on the right side of each plot show the time integrated and volumetrically constrained distributions of temperatures within zircon saturation for the three shown examples.

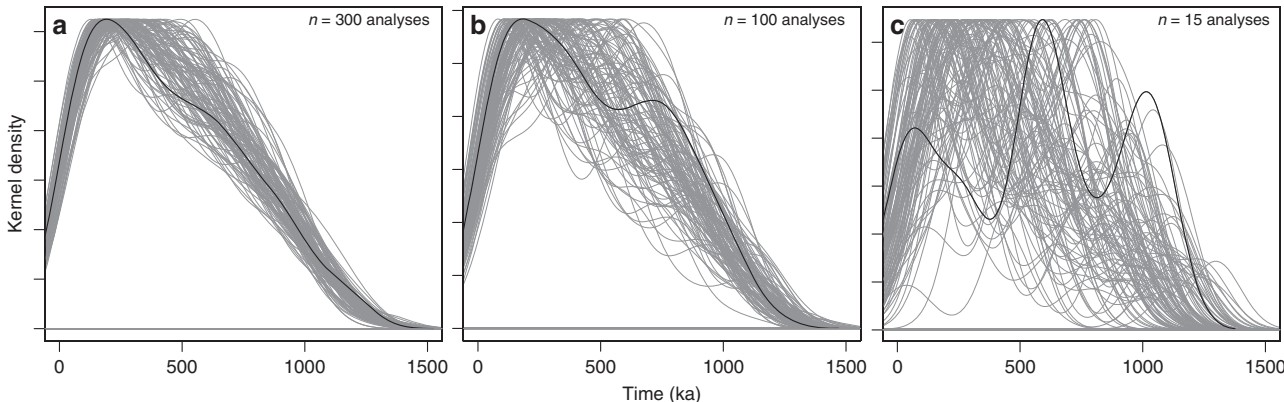

**Fig. 5 Example of synthetic zircon age populations computed from thermal modeling.** The kernel density estimate of the zircon age distribution is shown over the duration of the magma injection episode of 1.5 Ma for an incrementally-built intrusion with a magma flux of $9.1 \times 10^{-6}\ km^3\ km^{-2}\ yr^{-1}$, using a sill radius of 10 km and andesitic recharge magma[45]. The modeled zircon distribution was randomly sampled 100 times (gray curves) with (**a**) 300 measured synthetic zircon ages, (**b**) 100 analyzed zircon ages and (**c**) 15 synthetic zircon ages. Note the spikiness of the synthetic zircon distribution in (**c**), which is resulting from under-sampling of the distribution. The black curve in each plot represents a randomly selected distribution to illustrate the impact of under-sampling on the distribution shape. The bandwidth was set to 100 ka.

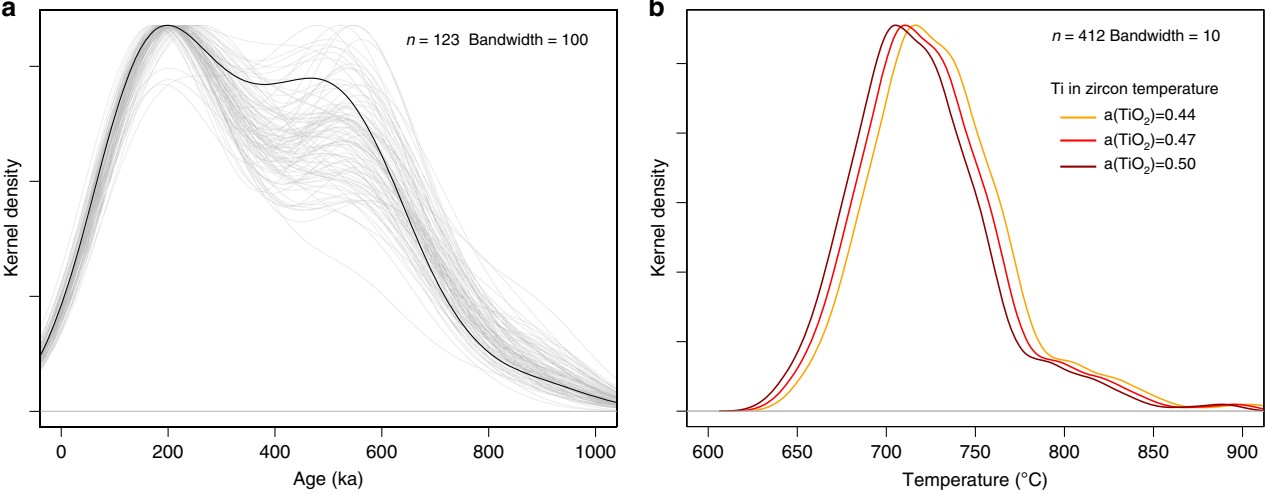

**Fig. 6 Natural zircon ages and Ti-in zircon temperatures. a** Combined zircon age population for all studied eruptions (i.e. Upper Toluca Pumice, Middle Toluca Pumice, Block and Ash Flow 28, White Quarry Pyroclastic Flow). The population has been resampled to account for oversampling of young ages by U-Th dating (Methods section) and to calculate the 2sigma zircon age span with 95% confidence interval in order to compare synthetic and natural zircon populations. Gray lines indicate individual random samples of the population with n=123 sampled zircon analyses. **b** Kernel density estimate of Ti in zircon temperatures of the combined zircon population for the 4 eruptions. The activity of Ti used in the calculations was evaluated by calculating the chemical affinity of rutile saturation using rhyolite-MELTS[67] and varied between 0.44 (orange line), 0.47 (red line) and 0.50 (dark red line) for typical dacite and andesite compositions from Nevado de Toluca[44]. The number of temperature estimates from zircon Ti contents (*n*) is shown on the top right of the diagram.

with 95% confidence and to constrain the time-integrated thermal evolution of the upper crustal plumbing system using Ti in zircon (Fig. 6). In order to assess the uncertainty of the combined U-Th and U-Pb total age spread (Fig. 6a; $n = 123$ analyses), we

bootstrap (1000 repetitions) the population and calculate the standard deviation of the total age spread for the 1000 resampled populations. This shows that the true 2σ age range of the natural population lies within the interval 380–470 ka with 95% confidence

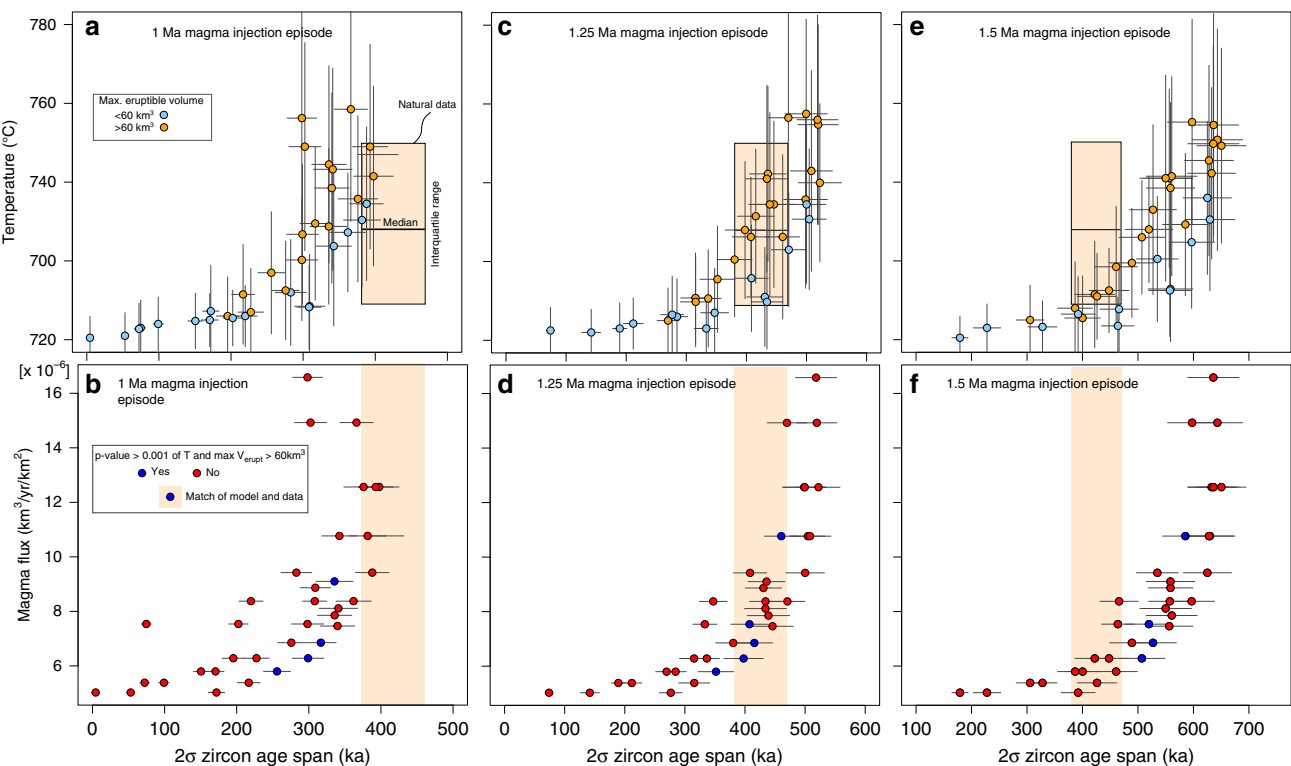

**Fig. 7 Relationship of temperature, zircon age span, and magma flux in model and natural data.** Circles show median of the temperature distribution within zircon saturation and the average 2σ zircon age span for individual models. Vertical gray error bars indicate the interquartile range of model temperatures and the horizontal bars the 95% confidence interval of the 2σ zircon age spans. Light blue color coding specifies a maximum eruptible magma volume <60 km³ and orange >60 km³ in the models. Natural 2σ zircon age span, median, and interquartile range of Ti in zircon temperatures are shown as tan box. Blue color coding of symbols in the lower subpanels indicates a temperature comparison between model and natural data with p-value > 0.001 and maximum eruptible volume in the model >60 km³, while red color indicates vice versa. **a** Temperature versus 2σ zircon age span for 1 Ma lasting magma injection episode. **b** Magma flux and 2σ zircon age span for 1 Ma lasting magma injection episode. **c** Temperature against 2σ age span for 1.25 Ma injection episode. **d** Magma flux versus 2σ zircon age span for 1.25 Ma lasting magma injection episode. **e** Temperature vs. 2σ zircon age span for 1.5 Ma lasting magma injection episode. **f** Magma flux plotted against 2σ zircon age span for 1.5 Ma lasting injection episode.

(Fig. 7). Ti in zircon crystallization temperatures were calculated for a range of Ti activities (Fig. 6b). For an intermediate Ti activity (Methods section; Supplementary Fig. 4) of 0.47, a mean zircon crystallization temperature of 726 °C with IQR of 699-745 °C is calculated (Fig. 6b). Considering that zircon from Nevado de Toluca records continuous melt presence in the plumbing system for at least 900 ka, this temperature distribution suggests that magma is primarily stored in a highly crystallized state for most of this time in an upper crustal plumbing system. For example, equilibrium crystallization of andesitic liquids indicates a crystallinity of 67 vol.% for a temperature of 725 °C[45]. Given typical whole rock Zr contents of 140 μg/g for Nevado de Toluca[39,44], which are in good agreement with most intermediate to silicic calc-alkaline magmas, Zr saturation temperatures <825 °C can be expected[45]. Typical eruptive temperatures of 820-960 °C for andesite and dacites from Nevado de Toluca[38,44] are close to, but generally in excess of zircon saturation, suggesting that the eruptive products are mixtures of hot recharge pulses and remobilized crystal-rich magmas. In order to sample the same overall zircon population in various temporally separated eruptions, melts and zircon crystals must have been efficiently transferred between various regions of the reservoir. Such a process is consistent with mechanical mixing related to the injection of hot magma and gas into a highly crystalline reservoir, triggering convective overturn of the system due to density instabilities[46–48]. In accordance with petrological evidence for influx of fresh magma batches prior to eruptions of

Nevado de Toluca[44], this observation implies that renewed magma injection into the subvolcanic reservoir is a mechanism required for the activation of this system.

**Long-term magma fluxes.** We compare the measured and calculated range of zircon crystallization ages and Ti-in-zircon temperatures with the calculated temperature evolution to estimate the average rate of magma input into Nevado de Toluca's plumbing system (Fig. 7). In order to ensure comparability between the model and natural data, the calculation of zircon age spans was carried out the same way and temperatures are compared over the same interval. The natural zircon temperature distribution indicates that crystallization extends to about 650 °C (Fig. 6b), which is lower than the assumed solidus temperature of 690 °C used in the thermal modeling. We therefore compare only temperature distributions within the interval 690 °C to 825 °C in both the model and natural data. Zircon age spans versus temperature and model magma flux are shown for total duration of magma injection of 1.0 Ma, 1.25 Ma, and 1.5 Ma (Fig. 7). Magma injection over 1.0 Ma produces zircon crystallization age distribution that for the great majority span a shorter age range with respect to the measured one (Fig. 7a, b). Calculated and measured total spread in zircon crystallization ages are comparable for a selection of models run for longer time spans (Fig. 7c–f). Besides temperature and range of zircon crystallization age, we further

constrain the characteristic magma flux for Nevado de Toluca by comparing its volcanic output and the maximum volume of eruptible magma (i. e. magma with <50% crystallinity and all interstitial melt) accumulated in the different simulations. Based on geological reconstructions, Nevado de Toluca has produced an absolute minimum output volume of 60 km³ during its 1.5 Ma history[39].

We use three criteria to identify the simulations that best match our measurements: (1) The range of zircon crystallization age of natural and synthetic zircon populations must agree within their uncertainties; (2) The maximum eruptible volume in the numerical simulation must be greater than the minimum eruptive output of 60 km³; (3) Correspondence between Ti-in-zircon temperature distributions and temperatures of the modeled intrusions. In order to make this comparison quantitative, we use a two-sample Welch's t-test to compare distribution means. To reject the null hypothesis that the population means are equal, a significance level needs to be defined. Visual inspection of the natural temperature distribution compared to modeling results indicates that population means differing by more than 10 °C, which clearly do not reproduce the natural data, result in $p$-values on the order of $10^{-6}$ (Supplementary Fig. 6). On this base, we use a significance level $p$-value< 0.001 to reject the null hypothesis. A few numerical simulations with injection duration of 1.0 Ma produce temperature distributions and zircon age spans that compare favorably to the natural data, but produce eruptible volumes < 60 km³ (light blue symbols; Fig. 7a), these simulations will not be discussed further. Using our approach, we find that four numerical simulations with magma injection durations of 1.25 Ma best match the natural data (Fig. 7d). Of these, one simulation at relatively high magma flux of $10.8 \times 10^{-6}$ km³ km⁻² yr⁻¹ produces a maximum eruptible magma volume of 62 km³, which is very close to the minimum value of 60 km³ and therefore we exclude it. The three other simulations cluster around a narrow range of magma fluxes between $5.8 \times 10^{-6}$ and $7.5 \times 10^{-6}$ km³ km⁻² yr⁻¹ and produce more realistic volumes of potentially eruptible magma over the 1.25 Ma of magma injection (148–341 km³). These last are our best estimates for the long-term rate of magma input into the plumbing system of Nevado de Toluca. The measured range in zircon ages is a minimum estimate because of the difficulty to analyze the youngest rim in sectioned zircon and the relatively low proportion of the oldest zircon crystals sampled by an eruption (Fig. 5). However, an increase of the zircon age span and duration of magma injection in the model to 1.5 Ma yields very similar magma flux estimate between $6.3 \times 10^{-6}$ and $7.5 \times 10^{-6}$ km³ km⁻² yr⁻¹ (Fig. 7f). Our crustal magma flux estimates for Nevado de Toluca fall between geologically observed crustal intrusion rates of $1 \times 10^{-5}$ to $1 \times 10^{-6}$ km³ km⁻² yr⁻¹ (Lipman and Bachmann, 2015; Dimalanta et al., 2002) and are similar to input rates estimated by numerical simulations of the incremental assembly of intermediate sized, relatively long-lived magmatic systems [33]. Our approach is principally applicable to any volcanic system in which zircon crystallizes, but care should be taken when directly extrapolating the results presented in Fig. 7 to determine crustal magma fluxes for other volcanoes. In particular, variables used in the thermal simulations, such as recharge magma temperature, intrusion depth, and initial geothermal gradient, should be similar to correctly estimate magma fluxes in other systems using the results we present here. Further systematic investigation of the parameter space will, however, allow us to directly apply our method to any volcanic systems for which zircon age and trace element data are available.

Our best estimate for upper crustal magma fluxes can be used to constrain the extrusive:intrusive (E:I) ratio of Nevado de Toluca volcano. Total volumes between 2277 and 2960 km³ of

magma have been injected in the numerical simulations that best-fit the natural zircon data. Considering an uncertainty of 50% on the absolute minimum estimate for the eruptive output[39], between 60 and 90 km³ are a reasonable approximation for the total eruptive output, yielding E:I ratios between 0.02 and 0.04 for Nevado de Toluca volcano. Estimates of the E:I derived by various methods[35] are subjected to considerable uncertainty and typically range between 1:1 and 1:34. Our calculations indicate that even when considering large uncertainties on the volume of erupted magma, only a few percent of the injected volume ever erupts, at least in system similar to Nevado de Toluca. Such low E:I imply that individual upper crustal magma bodies can produce both volcanic eruptions and hydrothermal ore deposits (e.g., of porphyry copper-type). Continuous melt presence for prolonged timescales, only marginally disturbed by eruptions in terms of total volume, will result in magma crystallization and degassing over protracted periods of time, which can explain the formation of porphyry copper deposits without the involvement of excessive metal enrichment in the degassing magma[49].

**Current state of the magmatic system and eruptive potential.** Our results provide information on the temporal evolution, current state, and eruptive potential of Nevado de Toluca volcano (Fig. 8). The time evolution of the volume fraction of melt with temperature >812 °C, equivalent to magma with < 50% crystallinity[45], which is corresponding to the typical rheological transition from the non-eruptible to eruptible state of silicic magmas[11], is shown for a numerical simulation that matches the natural data in Fig. 8a. These calculations demonstrate that volume fractions of magma above typical eruptive temperatures for Nevado de Toluca[38,44] are decreasing with time and are only present in direct association with recharge events over short time intervals throughout the entire history of the reservoir. This observation is consistent with petrological evidence for magma mixing and mingling of different crystal populations prior to Plinian eruptions from the volcano[44]. Zircon temperature distributions, which are much lower than eruptive temperatures of >820 °C for Nevado de Toluca, suggest that zircon (and melt) were remobilized from the non-eruptible portion of the system (i.e. at crystallinities >50% and temperatures <812 °C) during forceful intrusion of recharge magmas and hybridization with the resident mush. Figure 8 shows that the first 500 ka since the onset of magma recharge in the simulation are characterized by rapid cooling of single injection pulses, yielding high fractions of potentially eruptible magma relative to the total reservoir volume. After this time period, volume fractions of eruptible magma decrease below 0.2 at about 900 ka since the onset of injection, as more low melt fraction magma accumulates. This suggests that the E:I ratio of the magma reservoir decreases with time. The thermal and mechanical evolution of the crust might further decrease E:I, as increasing wall-rock temperature will favor the relaxation of overpressures generated by magma injection, thus decreasing the likelihood of eruption[50]. As shown by the simulation, eruptible magma is only present in the reservoir for timescales of years to centuries in direct association with a fresh recharge pulse (inset – Fig. 8a). This indicates that volcanic eruptions at Nevado de Toluca can only be initiated by mixing or mingling processes of fresh and resident magma. The short time window of eruptible magma presence in association with magma recharge is in good agreement with diffusion timescales and existing petrological evidence for the interaction between evolved and more mafic melts[44].

We now compare the total volume of magma that was available in the plumbing system over the history of Nevado de Toluca

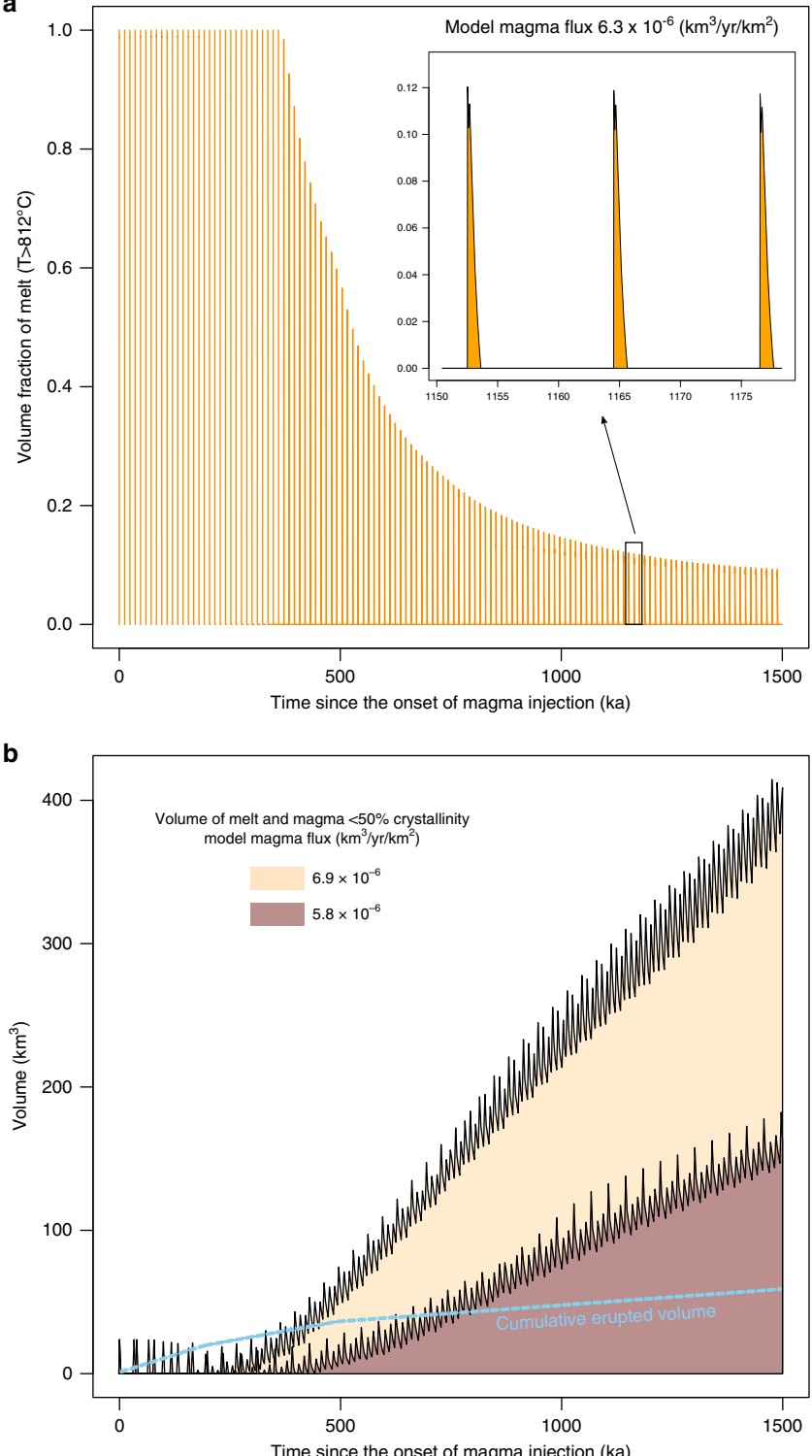

**Fig. 8 Time evolution of eruptible magma volumes. a** Time since the onset of magma injection (ka) versus volume fraction of melt with $T > 812\,°C$ (i.e. corresponding to a melt fraction >50 vol.%, which is commonly assumed as a threshold for magma eruptibility[11]) calculated for a model with magma flux of $6.3 \times 10^{-6}\ km^3\ yr^{-1}\ km^{-2}$ that matches the natural zircon age population. Note that the volume fraction of eruptible magma decreases with time. The inset shows a close up view of the presence of eruptible magma in the system, showing that the latter is only available for limited amounts of time and only in direct association with recharge events. **b** Cumulative volume of melt and magma with less than 50% crystallinity for magma fluxes of $6.9 \times 10^{-6}\ km^3\ yr^{-1}\ km^{-2}$ (tan area) and $5.8 \times 10^{-6}\ km^3\ yr^{-1}\ km^{-2}$ (red). Light blue dashed curve shows the minimum erupted volume from Nevado de Toluca volcano[39] for comparison. The simulations show that once a magmatic system reaches thermal maturity (i.e. melt accumulates following recharge pulses), large volumes of potentially eruptible magma can build-up in the upper crust. For the simulations shown here, which match the natural zircon age population, thermal maturity is reached after about 500 ka (red curve) and 400 ka (tan curve) since the onset of magma injection. Accumulated magma volumes after 1.5 Ma of injection in both scenarios are greater than the eruptive output of Nevado de Toluca.

with the total amount of erupted magma. The temporal evolution of the melt volume (km$^3$) is shown for numerical simulations with magma flux of $5.8 \times 10^{-6}$ and $6.9 \times 10^{-6}$ km$^3$ km$^{-2}$ y$^{-1}$, matching the zircon age and trace element measurements (Fig. 8b). Maximum volumes between 182 and 415 km$^3$ have accumulated after 1.5 Ma of magma injection, which is the total duration of volcanic activity observed at Nevado de Toluca[39]. Subtracting the minimum erupted output of 60 km$^3$ from this maximum value, a melt volume of 355 km$^3$ may be residing beneath the volcano today. It is worth noting that a volume of 350 km$^3$ was estimated for a magma chamber associated with the UTP eruption[51], considering a pressure range of 150 to 200 MPa, water-saturated conditions, decompression, and bubble growth conditions following the model of ref. [52]. The complete evacuation of this maximum estimated volume (355 km$^3$) in a single volcanic eruption is unlikely as it would require large volumes of recharge magma (Fig. 8a). Alternatively, interstitial melt segregation processes or reactive melt flow may assemble rhyolitic melts into crystal-poor melt rich eruptible pockets and feed large eruptions[15,53–55]. However, crystal-poor rhyolites with very low Ba and Eu contents have not been erupted by Nevado de Toluca over the last 1.5 Ma, indicating that such processes, even if they would occur, may not mobilize sufficient melt to result in eruptions at this volcano. Nevertheless, we propose that our estimate of the amount of eruptible magma present within the system could be compared with geophysical data and serve to estimate the size of the largest possible eruption that Nevado de Toluca would produce, which is an important parameter for volcanic hazard assessment.

Our findings show that eruptions at Nevado de Toluca occur following transient recharge pulses into a low melt fraction reservoir, on timescales comparable to human lifespans (Fig. 8a). Additionally, our calculations suggest that the upper crustal reservoir of this volcano is thermally mature and able to rapidly reawaken would magma supply from depth resumes. Our approach can be generalized to any volcanic system crystallizing zircon and can provide essential information for the assessment of volcanic hazard in dormant systems.

## Methods

**Zircon separation and imaging**. Sampling locations are described in Supplementary Table 1. Zircon crystals were separated from juvenile pumice or block clasts of the four studied eruptions. For each sample, composite clasts were crushed using a hydraulic press or steel jaw-crusher and sieved to obtain a size fraction <500 μm. The samples were further concentrated by using standard density and magnetic separation techniques including Wilfley table, hand magnet, Frantz magnetic separator, and heavy liquids separation. Zircon crystals in various size fractions were handpicked in ethanol under a binocular microscope. Crystals were mounted in epoxy resin, polished to expose the crystal cores by removing ~50 μm of thickness from the grains and coated with a 20 μm thick layer of carbon to ensure electrical conductivity before imaging with electron beam techniques. We used a cold cathode (CITL 8200 Mk 5-1) mounted on a petrological microscope, as well as a scanning electron microscope (JEOL JSM7001F), both housed at the University of Geneva, to reveal internal cathodoluminescence (CL) zonation textures of zircon (Supplementary Fig. 1).

**Trace element analysis**. Zircon was analyzed for major and trace elements by laser ablation-inductively coupled plasma mass spectrometry (LA-ICP-MS). We used an Element XF sector-field ICP-MS interfaced to a Resolution 193 nm ArF excimer laser ablation system housed at the Institute of Earth Sciences, University of Lausanne. Prior to each session, the instrument was tuned in linear scan mode on NIST-SRM 612 silicate glass. Oxide generation was mitigated to be <0.07% for $^{232}ThO^+/^{232}Th^+$ (with O referring to $^{16}O$). SRM 612 was also measured during the analytical session twice every 16 zircon analysis to correct for drift of the instrument. For sample analysis, a 10 Hz repetition rate was used and laser output energy was between 3-6 J cm$^{-2}$. The ablation spot size was varied between 24, 30, 38, and 50 μm depending on the size of zircon. A typical analysis consisted of 80 s background measurement followed by 35 s of laser ablation. We analyzed 229 individual zircon crystals, of which typically the cores and rims were analyzed for trace elements. The raw data were processed and transformed into elemental contents using the MATLAB based software package SILLS[56]. Zircon stoichiometric SiO$_2$

contents of 32.78 wt.% were used for internal standardization. We do not report measurements that showed elevated contents in Ca, Ba, Sr or Al, in order to exclude analysis subjected to contamination by inclusions or chemical alteration.

**U-Th geochronology**. U-Th zircon analyses were carried out by using a CAMECA ims 1280-HR ion probe at Heidelberg University (Germany), following the protocol described in ref. [57]. We used a mass-filtered O$^-$ primary ion beam with a current of ~50 nA focused to an analysis beam size of ~40 μm in diameter. A mass resolution power (m/mΔ) of ~5000 was achieved by the acceleration of secondary ions with 10 keV at an energy window of 50 eV using the narrow entrance and exit slit settings. The actinide oxides $^{230}ThO^+$, $^{232}ThO^+$, and $^{238}UO^+$, as well as the zirconium species $^{90}Zr_2O_4^+$ were simultaneous detected using electronmultipliers (EM) and Faraday cups (FC). To correct for background intensities, we subtracted the averaged intensities obtained on masses 244.03 and 246.3 amu from the $^{230}ThO^+$ intensities. Different fragments of the secular equilibrium reference zircon AS3 were analyzed to determine the U-Th relative sensitivity factor (RSF) from radiogenic $^{206}Pb/^{208}Pb$. Using this RSF calibration, AS3 reference zircon was then analyzed independently to assess accuracy. The weighted average for AS3 from three analytical sessions was ($^{230}Th/^{238}U$) = 1.006 ± 0.005 (1σ, MSWD = 0.85, n = 14). Background-subtracted $^{230}Th/^{238}U$ and $^{238}U/^{232}Th$ ratios for the unknowns were corrected using the RSF obtained from the calibration data, and the activity ratios were calculated[58]. We used the two-point isochron approach[59] to calculate $^{230}Th/^{238}U$ zircon model ages from the slope of the activity ratios and U-Th whole-rock isotopic compositions. As the latter is unknown for Nevado de Toluca, we used the whole-rock composition of El Chichón in southern Mexico[60]. We tested the differences in zircon ages calculated using the $^{230}Th/^{238}U$ groundmass composition of ref. [61] and Upper Continental Crust[62] relative to the values obtained by using the El Chichón whole-rock composition. Model ages calculated using the Colima groundmass value are on average 7.4 ± 4.7 ka (2 standard deviations) older compared to ages using the El Chichón composition, while the ages calculated with the values from ref. [62] are on average slightly younger (0.09 ± 1.9 ka).

**U-Pb geochronology**. Zircon U-Pb dating was performed on the same samples as U-Th dating and trace element analysis in two consecutive sessions using LA-ICP-MS housed at the University of Lausanne, Switzerland. The analyses were carried out on an Element XR sector-field spectrometer coupled to a Resolution 193 nm ArF excimer laser ablation system. Prior to each session, the instrument was tuned by linear scans on NIST-SRM 612 glass with 10 Hz repetition rate, 80 μm beam size and 6 J/cm$^2$ energy density. Before analysis the zircon mounts were cleaned with vol. 5% HNO$_3$ and deionized H$_2$O. Further, the sample surface was cleaned by 10 laser pulses prior to each measurement. Sample analysis was carried out with 5 Hz repetition rate, 3 J/cm$^2$ energy density and variable 50 μm spot size. The analysis routine consisted of duplicate measurements of SRM 612 at the beginning and end of a sample analysis sequence, 10 sample analysis, and duplicate measurements of zircon GJ-1, which was used as internal standard to calculate the relative sensitivity factor and test the accuracy. Measurements consisted of 45 s background and 20 s laser ablation. The raw intensity data reduction and $^{206}Pb/^{238}U$ age calculations were done, using the software package LAMTRACE. As the samples studied in this eruption are relatively young, $^{207}Pb/^{235}U$ ratios are too low to constrain meaningful ages and are thus not reported. A small fraction of crystals (3%) showed crystallization ages of >30 Ma, clearly indicating xenocrystic origin of these grains. We only report zircon analysis in this study that showed homogeneous intensity signals in time, to exclude mixed ages subject to overgrowth of inherited xenocrystic cores. As $^{204}Pb$ counts were too low to employ a common lead correction, we followed the approach presented in ref. [63] and filtered out analysis that indicated elevated common lead contents based on Concordia diagrams (Supplementary Fig. 2). The $^{206}Pb/^{238}U$ ages were corrected for initial Th disequilibrium[63]. Error bars reported in this study account for propagation of analytical uncertainty and the reproducibility of the internal standardization on zircon GJ-1.

**Thermal and zircon crystallization modeling**. The temporal evolution of temperatures in the Earth crust resulting from pulsed magma injection was simulated by numerical modeling. We solved the two-dimensional axisymmetric formulation of the heat conduction equation, which can be written as

$$\rho c \frac{\partial T}{\partial t} = \frac{1}{r} \frac{\partial}{\partial r} \left( rk \frac{\partial T}{\partial r} \right) + \frac{\partial}{\partial z} \left( k \frac{\partial T}{\partial z} \right) + \rho L \frac{\partial X_c}{\partial t}, \quad (1)$$

where $T$ is the temperature, $t$ is the time, $r$ is the radial coordinate relative to the symmetry axis, $z$ is the depth, $k$ is the thermal conductivity, $L$ is the latent heat of crystallization, $\rho$ is the density, $c$ is the specific heat, and $X_c$ is the fraction of crystals in the magma. A list of modeling parameters used in this study is shown in Table 1. Equation 1 was solved on a numerical grid corresponding to a crustal section of either 15 × 15 km or 20 × 20 km using an explicit finite difference solver. Zero flux boundary conditions were employed on all lateral model sides except the surface, which was fixed at 8 °C. In all simulations, an initial geothermal gradient of 40 °C/km was used, based on heat flow measurements in the Toluca area[64]. Recharge magma compositions at Nevado de Toluca are a priori unknown, but the composition of peripheral monogenetic cones and the presence of mafic crystal cores in the dacites indicates basaltic andesite to andesitic compositions[39,44].

**Table 1 List of parameters used in thermal modeling of magma injection in the Earth crust.**

| Parameters used in thermal modeling | |
|---|---|
| Specific heat | 1000 J kg$^{-1}$ K$^{-1}$ |
| Thermal conductivity | T-dependent[a] |
| Latent heat | $3.13 \times 10^5$ J kg$^{-1}$ |
| Density | 2700 kg m$^{-3}$ |
| Initial geothermal gradient | 40 °C km$^{-1}$ |
| Surface temperature | 8 °C |
| Intrusion depth | 6.5 to 8.5 km |
| Tliquidus | 1000[b] or 1115[c] °C |
| Tsolidus | 690 °C |

(a) Average crust as in Whittington et al. [66]
(b) Andesite liquidus from Marxer and Ulmer [45]
(c) Andesite liquidus of Blatter and Carmichael [65]

Therefore, we ran numerical simulations in which an andesite at its liquidus temperature of 1000 °C[45] or a more mafic andesite with liquidus of 1115 °C[65] was injected. Both liquid lines of descent are based on petrological experiments and are consistent with the mineral phase assemblages observed at Nevado de Toluca[39,44]. In all simulations, the solidus temperature was assumed to be 690 °C based on extrapolation of experiments by ref. [45]. Latent heat release in the model is governed by the relation between melt fraction and temperature, which was implemented by fitting a polynomial function through the experimental data for each of the two model recharge magma compositions. We modeled magma intrusion as sill-shaped cylindrical bodies with a radius of either 5 or 10 km, dimensions that are equivalent to the radius of the volcano's base. To test the impact of magma intrusion depth, injections in the model were either fixed at 6.5 km or randomly varied between 6.5 and 8.5 km, consistent with petrological findings for Nevado de Toluca[39,44]. Both emplacement modes yielded overall similar results (Supplementary Fig. 3). In response to magma injection, crustal rocks were advected downwards in all numerical simulations. We varied the volumetric magma fluxes between $5 \times 10^{-6}$ and $1.7 \times 10^{-5}$ km$^3$/km$^2$/yr and the duration of the magma injection episode from 1 to 1.5 Ma based on the observed zircon age span and volcanic activity[37,39].

The evolution of temperature with time (T–t path) was tracked in 500 numerical nodes of the modeled cylindrical magma reservoirs to calculate synthetic zircon age populations. The cumulative fraction of zircon crystallizing (F$_{zr}$) is linked to temperature by the following relation[26]:

$$F_{zr} = a - b \cdot 10^4 e^{-10^4/T}, \qquad (2)$$

where $a$ and $b$ are constants that change in relation to the zircon saturation temperature range and $T$ is the temperature in Kelvin. Using a hypothetical reversed relation of F$_{zr}$ and temperature results in synthetic zircon age spans that differ by ca. 70 ka from calculations using the experimental relation of ref. [26]. In all model calculations, we used the relation presented in Eq. 2. Further, in all calculations a zircon saturation temperature range of 825–690 °C was used based on petrological experiments[45]. We tested the impact of changing zircon saturation temperature on the age span of synthetic zircon populations by adjusting the upper bound to 860 °C and 800 °C, which resulted in age spans with a relative difference of 9 ka and 25 ka to our preferred zircon saturation window, respectively (Supplementary Fig. 7). At each temperature for each node we prescribe the crystallization of an arbitrary number of zircon crystals that is proportional to the temperature derivative of Eq. 2. Thus, we intrinsically assume that the number of measurable zircon ages is related to the fraction of zircon crystallizing. The number of measurable zircon ages is then calculated by summation of all T–t path at each time step. We assume that all zircon above the solidus temperature at the moment of eruption can be sampled in an eruption, as the natural zircon temperature distributions extend to the solidus. The T–t path with the longest permanence at a temperature within zircon saturation (and above solidus at the time of eruption) provides the maximum possible span of zircon ages. The age span of the synthetic zircon population is finally calculated as a 2σ range, encompassing 95% of the data around the median value. To calculate the confidence interval, we sample with replacement 123 zircon ages (equivalent to the number of analyses) from the calculated age distribution and repeat this process 1000 times.

In order to compare the natural and synthetic zircon age distributions, we resampled the natural population. This is necessary, as only ages <350 ka can be obtained by U-Th disequilibria dating due to the attainment of secular equilibrium in the $^{238}$U decay chain. Combining the U-Th and U-Pb ages therefore biases the age population towards higher numbers for ages <350 ka. Considering that the number of U-Pb ages smaller than 350 ka is 63 and greater than 350 ka is 60, we bootstrap (i.e. resample) the population of 63 U-Th ages and 60 U-Pb ages 1000 times to calculate the 2σ age span of the natural population. Ti in zircon temperatures have been calculated from the LA-ICP-MS analysis[40]. We

constrained the activity of Ti and Si by calculating the rutile and quartz affinities with rhyolite-MELTS[67] as a function of temperature for typical andesite and dacite composition from Nevado de Toluca (Supplementary Fig. 4).

## Data availability
The data collected for this study are available in the electronic supplementary materials.

## Code availability
The thermal code that was used in this study is available from the corresponding author upon request. The R script used for zircon crystallization calculations is available in the supporting materials to this article.

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

## Acknowledgements

G.W. and L.C. received funding from the European Research Council (ERC) under the European Union's Horizon 2020 research and innovation program (Grant agreement No. 677493-FEVER). We thank Alexey Ulianov for expert support during LA-ICP-MS analysis. Thomas Sheldrake is acknowledged for discussion on statistical aspects of this manuscript.

## Author contributions

G.W. and L.C. conceived this study. G.W. wrote the thermal code and ran numerical simulations. Zircon crystallization modeling was performed by G.W. and L.C. U-Pb dating was done by G.W. and U-Th dating by G.W. and A.S. Field work was carried out by G.W., L.C., and J. L. A. The initial manuscript draft was prepared by G.W. All authors contributed to data interpretation and writing of the final manuscript.

## Competing interests

The authors declare no competing interests.

**Additional information**

