## [Peer Review File · Nature Communications]

REVIEWER COMMENTS

Reviewer #2 (Remarks to the Author):

This contribution is exciting and novel for its coupling of two areas of research that have impacted igneous petrology over the last two decades: 1) zircon "petrochronology" combining geochronology and trace element geochemistry to constrain the longevity and fine-scale geochemical evolution of silicic magma chambers, and 2) thermophysical modeling of the inception, growth, and thermal evolution of magma chambers due to repeated intrusive activity. These two areas have yielded enormous contributions, but they are very different approaches whose results are difficult to link due to the different scales encompassed by their data and models.

The paper is well written and logically organized. I did not find any fundamental flaws in the approach and the assumptions used for the modeling are reasonable. The geochronology and trace element data is high quality and were derived using established analytical techniques.

The results of this study will be a big help to zircon (and other accessory mineral) petrochronology in igneous systems. The rise of sensitive microanalytical instrumentation and the use of zircon crystallization ages and trace elements to determine the durations of magmatic evolution, melt evolution, and geothermometry over the last two decades has allowed a large number of researchers to study volcanic and plutonic systems from all tectonic settings. A limitation for these studies is that uncertainties on crystallization ages are usually too large to resolve short-timescale (millennial) changes. However, the geochronology is certainly precise enough to infer the total duration of zircon crystallization in a magma reservoir. Moreover, the petrologic information determined from trace elements in a tiny zircon have been difficult to project and infer the evolution and melt abundance of a voluminous magma chamber.

Noteworthy conclusions that are impactful and will be of considerable interest to petrologists and volcanologists include:

- The modeling shows that an arc volcano can reawaken from multi-millennial dormancy within only a few years as it contains large volumes of near-solidus magma. This conclusion has important implications for understanding volcanic hazards and is unique because it integrates multiple types of natural data (zircon dates + trace elements, volcanic output).
- The modeling provides an explanation for interpreting the puzzling diversity of compositions that is often shown by zircons whose geochronology indicates coeval crystallization. The result indicating that higher magma flux results in greater variance of zircon trace element characteristics will be important for future studies using trace elements in zircon. Past studies have inferred that increased recharge or mixing could be responsible for variable trace element compositions in zircons but independent support from geophysical models of magma chamber evolution has been lacking. There are now LOTS of zircon trace-element data in the literature, and researchers have struggled to interpret their significance with respect to magma chamber dynamics. This conclusion will help.
- The random-sample demonstration of the amount of analyses needed to adequately characterize crystallization peaks (Fig. 5) will resonate with the zircon geochronology community. The need for 100 analyses to characterize an age distribution will not be a welcome bit of information but sometimes the truth hurts.
- Only a few percent of magma injected into the system ever erupts and extrusive:intrusive ratios decrease over time. Eruptible magma is only present for years to centuries after a pulse of recharge.

Questions/comments/suggestion by Line:

Line 154: Provide your thoughts on what those independent constraints might be; which would be the best ones? What other observations would lend credence to the T-t spectra?

Line 165: Mention if the zircons are euhedral, subhedral or anhedral since you speculate about whether the youngest rims were resorbed during the recharge before eruption.

Line 186: If you have glass compositions with Zr concentrations, you can calculate an eruption temperature that is directly comparable to the zircon saturation temperature from the bulk composition(s).

Line 233: Here or somewhere else compare these flux rates to those that have been determined elsewhere.

Line 267: Does "magma injection" here mean the very first pulse that creates the magmatic system? Or is "injection" being used synonymously with "recharge?"

Weakness:

The applicability of this study to other volcanoes is unclear and an obvious question while reading this paper. The title implies that the model can be generalized to many volcanoes but the issue is not really addressed. Addressing the generality/applicability of the model would be an improvement to the paper. For example, can the model results depicted in Figure 7 be extrapolated to determine the flux rate at a volcano in the Cascades provided the duration of zircon crystallization had already been constrained? Is the model too specific to Nevado de Toluca for extrapolation of the model results?

Finally, in the interest of replicability, consider making the code freely available on a site such as Github.

Congratulations on a nice study. I hope to see this paper in publication.

Jorge Vazquez

Reviewer #3 (Remarks to the Author):

Yan Zhan (UIUC)

The authors estimated the magma injection rates to a shallow magma reservoir by comparing zircon chronological and geochemical measurements with thermodynamic modeling. The paper provides some valuable thoughts on the buildup of upper crustal magma reservoirs. The paper advocated a novel method which can provide a better constraint on the thermal evolution of a magma reservoir.

However, the authors need to make it clearer (too little information can be obtained by Line 63 alone) that what is the major difference in methods between this study and the previous studies (e.g., Tierney et al., 2016)? In other words, what is the dominant effect impacting the precision and accuracy of magma inputting rate estimation, such as, a larger sample number (since you combined the results from several eruptions), better numerical models, or more quantitative statistical techniques (e.g., bootstrap, Welch's t-test)? Therefore, this paper can provide more influential thinking in the field.

I would suggest at least a moderate revision before this paper can be published in Nature Communications.

- Other major comments:

In your model, you assumed "all zircons above the solidus temperature at the moment of eruption can be sampled." However, even above the solidus, the high viscosity of the magma due to crystallinity will make this part of the magma non-eruptible. Do you consider a "mechanical solidus" (i.e., the viscosity of the magma is low enough for eruption), since, in Line 258, you mentioned you calculate the fraction of melt >

812 degC? So, I am confused about which temperature are you used to calculate the zircon population distribution? Also, it will be valuable to do some sensitivity tests showing the effects of the chosen temperature window on the results of the zircon age distribution.

In Line 63, How you do you know the higher resolution by this method can also be a better solution? I noticed you mentioned some independent evidence later. Also, if you are using all the total spread of zircon ages with the data from the previous paper (e.g., Tierney et al., 2016), can you also make a more precise estimation of magma inputting rate?

- Some minor comments:

Line 96. Please make a comment here, whether and why a similar zircon trace element signature can ensure a more (or less) accurate estimation of magma injection rate.

Line 153. Please specify, at which wavelength or scale, the peak/gap is led by the lack of samples. Since the overall zircon age distribution, which is also a peak, is due to magmatic processes.

Line 210. Fig. 7c-f?

Fig. 1 The font size of the inserted plot is too small to read.

Figure 4. From the plots, it's hard to believe the orange lines are the average of those grey lines, especially for the figure (C). Maybe a larger population of grey lines are covered by the orange lines with black error bars. Maybe, a density spectrum against time plot may help to see the distribution of those grey lines.

Figure 7. I suggest the author highlight the idea that the hypothesis will be accepted or rejected if the tan boxes in the plots are overlapping the circles with which colors, by either adding statements in the figure caption or pointing out that on the plots. Also, it is not obvious that the plots in the same column show the same X Ma magma injection episode.

Response to Reviewers

Estimating the current size and state of subvolcanic magma reservoirs

Gregor Weber, Luca Caricchi, José L. Arce, and Axel K. Schmitt

Reviewer comments are shown in *italic* and our responses in **bold** font.

Comments by Reviewer #2 Jorge Vasquez:

This contribution is exciting and novel for its coupling of two areas of research that have impacted igneous petrology over the last two decades: 1) zircon “petrochronology” combining geochronology and trace element geochemistry to constrain the longevity and fine-scale geochemical evolution of silicic magma chambers, and 2) thermophysical modeling of the inception, growth, and thermal evolution of magma chambers due to repeated intrusive activity. These two areas have yielded enormous contributions, but they are very different approaches whose results are difficult to link due to the different scales encompassed by their data and models.

The paper is well written and logically organized. I did not find any fundamental flaws in the approach and the assumptions used for the modeling are reasonable. The geochronology and trace element data is high quality and were derived using established analytical techniques.

The results of this study will be a big help to zircon (and other accessory mineral) petrochronology in igneous systems. The rise of sensitive microanalytical instrumentation and the use of zircon crystallization ages and trace elements to determine the durations of magmatic evolution, melt evolution, and geothermometry over the last two decades has allowed a large number of researchers to study volcanic and plutonic systems from all tectonic settings. A limitation for these studies is that uncertainties on crystallization ages are usually too large to resolve short-timescale (millennial) changes. However, the geochronology is certainly precise enough to infer the total duration of zircon crystallization in a magma reservoir. Moreover, the petrologic information determined from trace elements in a tiny zircon have been difficult to project and infer the evolution and melt abundance of a voluminous magma chamber.

Noteworthy conclusions that are impactful and will be of considerable interest to petrologists and volcanologists include:

- The modeling shows that an arc volcano can reawaken from multi-millennial dormancy within only a few years as it contains large volumes of near-solidus magma. This conclusion has important implications for understanding volcanic hazards and is unique because it integrates multiple types of natural data (zircon dates + trace elements, volcanic output).*
- The modeling provides an explanation for interpreting the puzzling diversity of compositions that is often shown by zircons whose geochronology indicates coeval crystallization. The result indicating that higher magma flux results in greater variance of zircon trace element characteristics will be important for future studies using trace elements in zircon. Past studies have inferred that increased recharge or mixing could be responsible for variable trace element compositions in zircons but independent support from geophysical models of magma chamber evolution has been lacking. There are now LOTS of zircon trace-element data in the*

literature, and researchers have struggled to interpret their significance with respect to magma chamber dynamics. This conclusion will help.

- The random-sample demonstration of the amount of analyses needed to adequately characterize crystallization peaks (Fig. 5) will resonate with the zircon geochronology community. The need for 100 analyses to characterize an age distribution will not be a welcome bit of information but sometimes the truth hurts.*
- Only a few percent of magma injected into the system ever erupts and extrusive:intrusive ratios decrease over time. Eruptible magma is only present for years to centuries after a pulse of recharge.*

Questions/comments/suggestion by Line:

Line 154: Provide your thoughts on what those independent constraints might be; which would be the best ones? What other observations would lend credence to the T-t spectra?

We have added a statement in Lines 157-158 that to our mind equivalence between independent thermo-chronometers such as titanite and zircon may help supporting episodicity in T-t histories.

Line 165: Mention if the zircons are euhedral, subhedral or anhedral since you speculate about whether the youngest rims were resorbed during the recharge before eruption.

In the revised manuscript, we present more information on zircon shapes and discuss whether the observations are consistent with the lack of ages close to eruption age. (Lines 173-175).

Line 186: If you have glass compositions with Zr concentrations, you can calculate an eruption temperature that is directly comparable to the zircon saturation temperature from the bulk composition(s).

We did not analyze trace elements in the glasses as these are analytically challenging due to high vesicularity (samples: UTP, WQ, MTP) and crystalline groundmass (BAF). However, in the revised version of the manuscript, we discuss the impact of changing zircon saturation temperatures on our results at Lines 450-453.

Line 233: Here or somewhere else compare these flux rates to those that have been determined elsewhere.

We now compare our flux estimates to geological and previously modelled estimates at Lines 250-254 of the revised manuscript.

Line 267: Does “magma injection” here mean the very first pulse that creates the magmatic system? Or is “injection” being used synonymously with “recharge?”

Here magma injection and recharge were used synonymously. In order to avoid ambiguity, we changed the wording to “magma recharge”. (Line 292)

Weakness:

The applicability of this study to other volcanoes is unclear and an obvious question while reading this paper. The title implies that the model can be generalized to many volcanoes but the issue is not really addressed. Addressing the generality/applicability of the model would be an improvement to the paper. For example, can the model results depicted in Figure 7 be extrapolated to determine the flux rate at a volcano in the Cascades provided the duration of zircon crystallization had already been constrained? Is the model too specific to Nevado de Toluca for extrapolation of the model results?

Thank you for pointing this out. We fully agree that the paper benefits from addressing the applicability of our approach to other systems and present a discussion of this issue in Lines 254-261 of the revised manuscript.

In summary, the results presented in Fig. 7 can be directly extrapolated to other systems, if the variables used in the thermal model are similar for the system under investigation. In particular, care should be taken that the temperature of the recharge magma (1000-1100°C), initial geothermal gradient (40°C/km) and intrusion depth (~5 km) are equivalent. While these conditions match a wide range of intermediate to silicic systems, extending the parameter space of the thermal model in future efforts will allow to apply this approach directly to a larger number of systems.

Finally, in the interest of replicability, consider making the code freely available on a site such as Github.

We have added the zircon crystallization code and thermal model outputs as part of the supplementary materials in order to ensure the replicability of our results. The thermal code is currently not optimized for wide dispersion but available upon request from the corresponding author. We are currently working to provide a widely applicable and user-friendly version of our model that will be freely available to the community.

Congratulations on a nice study. I hope to see this paper in publication.

Thank you very much for this assessment.

Comments by Reviewer #3 Yan Zhan:

The authors estimated the magma injection rates to a shallow magma reservoir by comparing zircon chronological and geochemical measurements with thermodynamic modeling. The paper provides some valuable thoughts on the buildup of upper crustal magma reservoirs. The paper advocated a novel method which can provide a better constraint on the thermal evolution of a magma reservoir.

However, the authors need to make it clearer (too little information can be obtained by Line 63 alone) that what is the major difference in methods between this study and the previous studies (e.g., Tierney et al., 2016)? In other words, what is the dominant effect impacting the precision and accuracy of magma inputting rate estimation, such as, a larger sample number (since you combined the results from several eruptions), better numerical models, or more quantitative statistical techniques (e.g., bootstrap, Welch's t-test)? Therefore, this paper can provide more influential thinking in the field.

Thank you for pointing this out. We have added further discussion to make clearer what the difference between our method and previous work is (Lines 63-68). The main difference to other models is the comparison between model and natural data. While previous studies used parameters describing the shape of the zircon age population to compare the model to natural data (Caricchi et al., 2014, 2016; Tierney et al., 2016), we use the 2 sigma value of the total duration of zircon crystallization, together with Ti-in-zircon temperature distributions and eruptive output for comparison. Increasing the number of independent constraints to compare models and natural data reduces the number of degrees of freedom and therefore constrains the magma input rates at higher resolution (less than a factor two with respect to an order of magnitude in previous model).

I would suggest at least a moderate revision before this paper can be published in Nature Communications.

- Other major comments:

In your model, you assumed "all zircons above the solidus temperature at the moment of eruption can be sampled." However, even above the solidus, the high viscosity of the magma due to crystallinity will make this part of the magma non-eruptible. Do you consider a "mechanical solidus" (i.e., the viscosity of the magma is low enough for eruption), since, in Line 258, you mentioned you calculate the fraction of melt > 812 degC? So, I am confused about which temperature are you used to calculate the zircon population distribution?

The synthetic age population was calculated for all zircons above the solidus temperature as stated in Lines 448-450. In order to clarify the rationale behind this, we have added a statement that the natural zircon age population also extends to the solidus temperature (Line 458).

The mentioned 812°C mark the point in the implemented melt fraction-temperature relation in the thermal model at which a crystallinity of 50% is reached, which is widely assumed to be the transition between the eruptible and non-eruptible state of magmas (e.g. Marsh, 1981). While the temperatures (typically >812°C) and crystallinities (typically ~40%) of eruptive products from Nevado de Toluca are consistent with this behavior, the zircon temperature spectra are much lower indicating that most of the

crystals are entrained in the eruptible portion by a remobilization mechanism probably during magma recharge.

In order to clarify this in the manuscript, we added further discussion in Lines 281-282 and Lines 288-291.

Also, it will be valuable to do some sensitivity tests showing the effects of the chosen temperature window on the results of the zircon age distribution.

As suggested by the Reviewer, we have tested the impact of changing the zircon saturation temperature window. Calculations with zircon saturation temperatures between 860-690°C and 800-690°C result in synthetic zircon age spans that differ from our preferred values (i.e. 825-690°C) by 9 ka and 25 ka, respectively. Hence, these differences are within the uncertainty envelope (tan boxes) of the age span given in Fig. 7. We discuss these findings in Lines 446-453 of the revised manuscript and present a new supplementary Figure 7 to illustrate this.

In Line 63, How do you know the higher resolution by this method can also be a better solution? I noticed you mentioned some independent evidence later.

If the higher precision of our method is also more accurate can only be tested by independent means. As the reviewer understood, we present some independent corroborating evidence in Lines 312-314, but further comparison (e.g. geophysical data) of this kind is not possible for Nevado de Toluca at present. However, the application of our methods to volcanic systems for which geophysical studies exist (e.g. Mt St Helens) would provide additional means to assess the accuracy of our method (comparing for example the total size of the thermal anomaly obtained by geophysical methods and estimated from our thermal models). Nevertheless, as we state in Lines 63-68 of the revised manuscript, we constrain our numerical model by 3 observational parameters rather than a single one used in previous work, which probably translates into higher accuracy.

Also, if you are using all the total spread of zircon ages with the data from the previous paper (e.g., Tierney et al., 2016), can you also make a more precise estimation of magma inputting rate?

Before considering to apply our technique to a different dataset, a number of parameters (e.g. initial geotherm, age span and recharge magma temperature) should be assessed in order to ensure to applicability of the numerical model to the specific system.

The dataset presented in Tierney et al., (2016) represents a case that is not directly applicable to the numerical simulations in our study, as parameters such as initial geotherm and total duration of magmatism differ from our model. Keeping these limitations in mind, 2 of the Dome samples in this study show comparable 2σ zircon age spans to our simulations (Chanka: 405 ka and Chascon 415 ka). Combining these estimates with the respective median Ti-in-zircon temperatures of 702°C and 696°C, indicates that the recharge rates are roughly between 4×10^{-6} to 6×10^{-6} km³/km²/yr (Fig. 7f). This is on the same order of magnitude but more precise compared to the estimate of Tierney et al., (2016).

A full assessment of this or other datasets would require to run a large number of additional simulations and is beyond the scope of this article. We therefore do not discuss the above mentioned calculations in the main text, but focus the discussion on the applicability of our method to other systems and compare our findings to previous magma flux estimates in Lines 250-261 of the revised manuscript.

- Some minor comments:

Line 96. Please make a comment here, whether and why a similar zircon trace element signature can ensure a more (or less) accurate estimation of magma injection rate.

The similarity in zircon trace element abundances allows us to combine the zircon populations from different eruptions as this indicates that the crystals originate from a common reservoir and by similar processes. If or how a heterogeneous or homogeneous zircon trace element population would impact on the magma flux estimate, we cannot tell from our study. To clarify what the significance of the similarity in zircon trace elements is, we added a statement in Lines 98-99 of the revised manuscript.

Line 153. Please specify, at which wavelength or scale, the peak/gap is led by the lack of samples. Since the overall zircon age distribution, which is also a peak, is due to magmatic processes.

We are not entirely sure about this comment. There is no characteristic wavelength or scale in the under-sampling of the continuous uni-modal distribution that creates the peaks and gaps in the synthetic zircon age population. For natural zircon age populations, both the possible effect of under-sampling for small sample sizes and differences in analytical uncertainty should be considered when assigning geological significance to peaks and gaps, but we cannot specify a scale or wavelength here.

Line 210. Fig. 7c-f?

Thanks for spotting this. It has been corrected.

Fig. 1 The font size of the inserted plot is too small to read.

The font size has been increased.

Figure 4. From the plots, it's hard to believe the orange lines are the average of those grey lines, especially for the figure (C). Maybe a larger population of grey lines are covered by the orange lines with black error bars. Maybe, a density spectrum against time plot may help to see the distribution of those grey lines.

We now state more clearly in the revised caption to Fig. 4 that the grey lines are just a small subsample of the entire T-t paths of a particular numerical simulation and that the apparent high density of grey lines at high temperatures results from the large thermal oscillations that individual T-t tracers close to the injection site undergo during their evolution.

Figure 7. I suggest the author highlight the idea that the hypothesis will be accepted or rejected if the tan boxes in the plots are overlapping the circles with which colors, by either adding statements in the figure caption or pointing out that on the plots. Also, it is not obvious

that the plots in the same column show the same X Ma magma injection episode.

Following the Reviewers suggestion, we have added an additional entry to the legend in Fig. 7b explaining that a match between model and natural data is achieved, if the tan boxes overlap with dark blue points. We now also indicate the duration of the magma injection episode in Figs. 7b, d and f.

REVIEWERS' COMMENTS

Reviewer #3 (Remarks to the Author):

Thank you for addressing my questions and comments! I have no more comments.